

# Detection of malicious nodes based on consortium blockchain

Song Luo, Lianghai Lai, Tan Hu and Xin Hu

College of Computer Science and Engineering, Chongqing University of Technology, Chongqing, China

## ABSTRACT

With the development of technology, more and more devices are connected to the Internet. According to statistics, Internet of Things (IoT) devices have reached tens of billions of units, which forms a massive Internet of Things system. Social Internet of Things (SIoT) is an essential extension of the IoT system. Because of the heterogeneity present in the SIoT system and the limited resources available, it is facing increasing security issues, which hinders the interaction of SIoT information. Consortium chain combined with the trust problem in SIoT systems has gradually become an important goal to improve the security of SIoT data interaction. Detection of malicious nodes is one of the key points to solve the trust problem. In this article, we focus on the consortium chain network. According to the information characteristics of nodes on the consortium chain, it can be analyzed that the SIoT malicious node detection combined with the consortium chain network should have the privacy protection, subjectivity, uncertainty, lightweight, dynamic timeliness and so on. In response to the features above and the concerns of existing malicious node detection methods, we propose an algorithm based on inter-block delay. We employ unsupervised clustering algorithms, including K-means and DBSCAN, to analyze and compare the data set intercepted from the consortium chain. The results indicate that DBSCAN exhibits the best clustering performance. Finally, we transmit the acquired data onto the chain. We conclude that the proposed algorithm is highly effective in detecting malicious nodes on the combination of SIoT and consortium chain networks.

## INTRODUCTION

In recent years, the Internet of Things (IoT) has evolved rapidly (*Roopa et al., 2019*) in the IoT paradigm, taking social characteristics into account, giving rise to a new concept of social networks consisting of smart objects, services, or both, known as Social Internet of Things (SIoT). SIoT is an extended area of IoT that emphasizes the application of IoT technologies to social and interpersonal relationships. The focus of SIoT is primarily on combining IoT technologies with social and interpersonal interactions to improve the quality of people's lives, enhance the integrity of society, and solve social problems. *Atzori, Iera & Morabito (2011)* benefits human users primarily through service discovery and combination, laying the foundation for autonomous interactions between objects. Incorporating this social network concept into the IoT paradigm can effectively address

Corresponding author
Lianghai Lai,
lailianghai@foxmail.com

the challenges faced by the IoT ecosystem. At the same time, with the development of the information society, the value of its information is gradually appreciating (*Mahdavinejad et al., 2018*). The role of traditional security technologies is becoming smaller and smaller, and it is more likely that SIoT system devices will be subject to security threats (*Deogirikar & Vidhate, 2017*). The data produced by SIoT system devices can be easily attacked, leading to falsehoods and inconsistencies in information. In more severe cases, threatening the privacy and information security of SIoT users (*Abdelghani et al., 2019*). In addition, one of the characteristics of SIoT is that it has distributed qualities, and SIoT information flow and security technologies should be more inclined to distributed approaches (*Roman, Zhou & Lopez, 2013*). Therefore, integrity, anonymity, information protection, and non-repudiation should be considered when designing IoT security solutions. Applying blockchain technology to the Internet of Things can solve part of the security problems (*Kumar & Sharma, 2022*). Blockchain can provide open, autonomous, decentralized, anonymous, and information-non-tamperable environments for IoT systems. Through the security benefits of blockchain, blockchain can be a reliable database for storing and sharing information on SIoT systems.

SIoT systems require secure and trustworthy environments, *Din et al. (2018)* will trust management systems based on machine learning or artificial intelligence or any mathematical calculations to compute the trust scores or reputation of SIoT nodes. The information for these calculations can come from the data collected by the SIoT devices or from the reputation scores calculated by different trust management techniques, and the reputation scores of the SIoT nodes can visually reflect the trustworthiness of the nodes (*Roopa et al., 2019*). Trust in SIoT by considering the following two points: subjective trust: (1) In subjective trust, direct quality of service trust is evaluated through the self-observation of the node, which is the self-observation of the node, *i.e.*, the observation of the node's historical behavior. (2) In objective trust, direct and indirect service quality trust is evaluated through feedback from other objects. In this article, we mainly consider subjective trust in services or transactions, *i.e.*, service quality trust is evaluated through the self-observation of nodes. After reviewing the literature, it is easy to conclude that distinguishing between malicious and non-malicious nodes by considering subjective trust is one of the key steps in solving the security problem of SIoT systems.

In this article, our research focuses on distinguishing between malicious and non-malicious nodes in SIoT systems combined with consortium blockchain. The main SIoT issues or challenges covered in this article are as follows:

1) Interaction security: increasing exposure to malicious objects and ensuring reliable interactions between SIoT objects.
2) Data privacy security: Data security and privacy protection are crucial issues in SIoT applications. A large amount of SIoT data and personal data is transmitted over the network, so appropriate measures must be used to ensure data security. For example, data encryption, anonymization, security auditing and monitoring, and traceability are ubiquitous security tools. Data security and privacy protection are vital factors for SIoT applications.

3) Data analytics and intelligence: This is also an essential part of SIoT. They analyze and extract from the data generated by IoT devices and sensors and make specific decision analyses. Data analytics and intelligence will continue to bring more innovation and value to SIoT applications.

4) Distributed environment: In a distributed network, there is no centralized entity, and each object creates various relationships to navigate the web to discover services and things. Distributed can improve SIoT processing efficiency as well as security.

5) User experience and design: user experience needs to be considered in SIoT to ensure that the user's needs are met and for the user to be able to use and enjoy the convenience of the new technology easily.

Consortium blockchain are a type of blockchain network, and consortium blockchain can be applied to almost any scenario that requires multi-party participation, privacy protection, efficient transactions, and trust mechanisms (*Dib et al., 2018*). Since 2017, the application of consortium blockchain technology in social IoT systems has also received increasing attention from international organizations and national governments, with the financial sector being one of the first industries to adopt consortium blockchain technology on a large scale (*Gongqiang, 2021*). One of the main features of the consortium blockchain is that members must be invited to join the network, which means that only authorized members can participate in the consensus process and access and validate transaction data, which provides a higher level of privacy and security for SIoT. The decentralized, traceable, and smart contract features of the consortium blockchain offer new possibilities for the security and trust building of the SIoT system. There are many connected devices in the SIoT system, and the security of these devices directly impacts the stability of the whole system. The decentralized features of the consortium blockchain enable it to establish a more secure data storage and interaction mechanism (*Lifang, 2019*), which helps to prevent malicious nodes from attacking and information tampering. In conclusion, a consortium blockchain is a form of a blockchain network with a high degree of authorization, privacy, security, and a focus on cooperation, which is suitable for a variety of application scenarios between enterprises and organizations, providing them with a more controlled way of sharing data and automating business processes.

This work demonstrates the algorithm's applicability in detecting malicious behavior. The research objective of this article is to be applied to SIoT nodes on the consortium blockchain to detect the presence of anomalies in the nodes. The SIoT nodes provide the data through the interactions between the IoT devices. After storing the data in the database, the data is intelligently analyzed to get the characteristics of the nodes and maintain the security of the system so that a small network of communities is formed, as shown in Fig. 1. In SIoT, smart service is one of its essential components. Transaction latency in smart service is a key performance metric that indicates the time taken from transaction initiation to transaction completion; thus, reducing transaction latency is essential to improve user experience, system efficiency, and quality of service. When a node conducts a transaction or service with another node, the algorithm proposed in this article can be carried out to determine whether the node has malicious or negative behaviors.

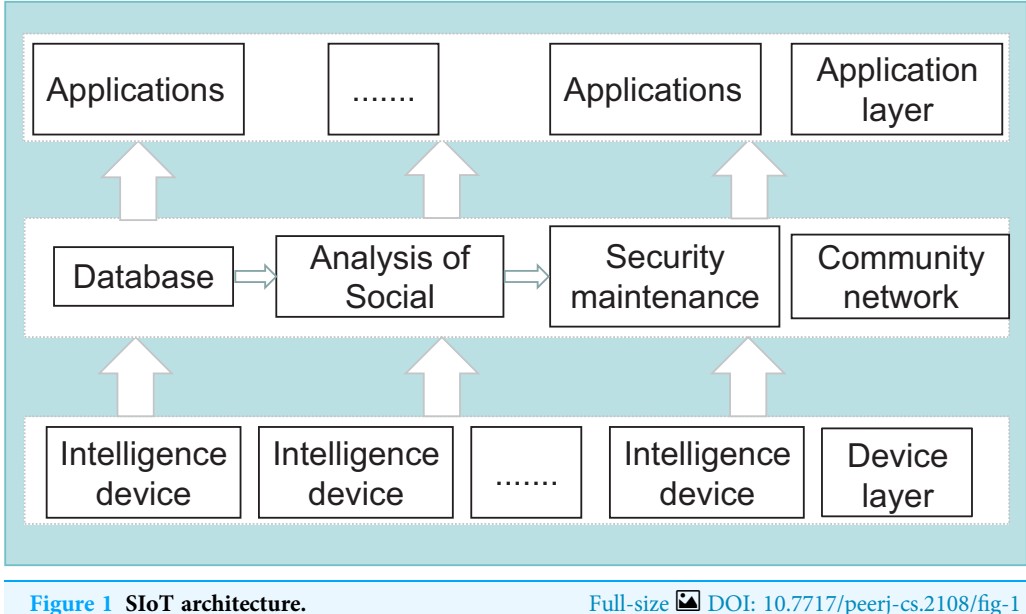

**Figure 1 SIoT architecture.**

Because processing delay and activity reflect the subjectivity of trust assessment and the heterogeneity of SIoT data, *i.e.*, the large amount and different information of SIoT data, a user with a slightly considerable processing delay in the system cannot be directly judged as a malicious node user. There is scientific evidence that many individuals in a social network provide much more accurate answers to complex questions than a single individual (*Jordan & Mitchell, 2015*). In the data information (especially transactions or services) generated by the SIoT system combined with the consortium blockchain, we have used machine learning techniques to help identify any patterns related to abnormal behavior. Machine learning clustering algorithms are used to partition the dataset into different categories according to a particular criterion of the features, such that the data in the same type has a great deal of similarity and the data in different categories has a great deal of difference. We obtained the processing latency and activity of each node by analyzing the block information on the blockchain. We used the unsupervised clustering algorithms DBSCAN K-means comparative analysis in machine learning to distinguish the malicious nodes in the SIoT system. The simulation experiments verify that the method is subjective, uncertain, lightweight, and dynamically time-sensitive for detecting malicious nodes. Our research proposes a methodology for detecting malicious nodes, but we acknowledge that our work is limited by specific types of attacks. The definition of a malicious node depends heavily on the attacker's ends and means, and our approach focuses on the types of delay or negative attacks in SIoT. Therefore, we only explore this type of malicious behaviour in this article, and different defence strategies may be required in other scenarios.

The main contribution of this article:

1) The malicious nodes are distinguished by detecting the historical processing intervals of SIoT nodes. This node detection technique is distinguished only by the historical

transaction records of the consortium blockchain and does not involve the specific transactions and node information of the specific nodes, which protects the privacy of the nodes;

2) A new processing delay algorithm is proposed, which records the interval between the most recent operation of a node compared to the history of operations of that node to obtain the delay value, and when the delay value is abnormal, then it indicates that the node may have an abnormal behavior in this time period;

3) We propose a subjective approach: We store the detection results in a consortium blockchain and make this data publicly available to all users on the chain. Users can view this data and use it to determine whether to perform a transaction or service on that node. For example, node A sees that node B has been identified as a malicious node in a previous transaction. However, node A can still trust node B to perform a transaction with node B. This demonstrates the subjective nature of the approach and improves the user experience of the SIoT service;

4) Comparative analysis using DBSCAN and K-means, K-means its computational complexity is relatively lower but does not incorporate the eigenvalue delay well into the detection. The situation in SIoT nodes is usually complex and varied, and the anomalies cannot be generalized. Through the results, it can be concluded that DBSCAN is the best algorithm in that context.

The remainder of this article is organized as follows, and related work is described in "Related Work". "Definition of Malicious Nodes and Introduction to Related Algorithms" discusses SIoT and blockchain technologies, and "Experiments and Results Analysis" presents simulation experiments and analysis of the final results. Our conclusions and future work are presented in "Conclusions".

## RELATED WORK

*Malekshahi Rad et al. (2020)* highlighted the history of SIoT from its origins to its evolution from IoT to SIoT. They identified some of the key characteristics of SIoT, which include usability, privacy, and security. They also explore future open problems, analyze large amounts of data to describe human dynamics and propose new relevant models.

Since SIoT systems combine many daily life objects from heterogeneous environments and the number of security issues that blockchain can address for SIoT systems is limited, some issues with the security of SIoT systems require trust management (*Blaze, Feigenbaum & Lacy, 1996*). The term "trust management" was first introduced, which they viewed as a separate aspect of security services in the network architecture, and defined that trust management provides a single way to determine and analyze credentials, security policies, and relationships (*Abdelghani et al., 2016*). Recently, there have been several reviews of trust management in SIoT, while concepts such as service discovery, relationship management, and service combination have not been addressed in SIoT settings. *Grandison & Sloman (2000)* defined trust as beliefs that are influenced by an individual's perceptions of certain key system characteristics. *Kini & Choobineh (1998)*

defined trust as A strong belief in an entity's ability to act reliably, safely, and securely in a given environment.

Our work is based on the SIoT environment on the consortium blockchain and focuses on the processing delay of malicious nodes in this environment. Malicious node detection is the key to trust management. The processing delay can be used as a source of trust for trust evaluation and provide a new solution for future SIoT trust evaluation. Since the data information provided by IoT and the scale of IoT is huge, the introduction of machine learning algorithms to deal with large-scale data is an essential tool for future research and development of IoT security (*Tahsien, Karimipour & Spachos, 2020*).

*Saini & Khari (2011)* differentiates between normal and malicious nodes for *ad hoc* networks and gives solutions for various malicious node intrusions. However, in it, no corresponding solution is given for delayed processing attacks on malicious nodes. *Dhingra, Jain & Jadon (2021)* used the simplified dataset CIDDS-001, which has five classes of labeled objects, two clustering algorithms, k-means, and DBSCAN, were used to process the dataset, and finally, the DBSCAN algorithm had the best clustering results. However, after a node is categorized as malicious, the non-malicious behavior of that node is also ignored, lacking the subjectivity and dynamic timeliness of the judgment. *Narayana & Midhunchakkaravarthy (2020)* (ICIRCA-2020) describes a trust assessment and block analysis-based model for the detection of malicious nodes in mobile *ad hoc* networks. Malicious nodes in the network are detected by analyzing the blocks generated by each node after transmitting data. Detecting malicious nodes in the network by analyzing the blocks generated by each node after transmitting data.

*O'Kane (2018)* performed direct k-means clustering and simultaneous means and PCA clustering on transactions with labels in Ether, where the accuracy of the simultaneous k-means and PCA clustering methods achieved an accuracy of 81.2%. However, in the feature selection, the Gini coefficients of block number and date are not taken into account as features because they are too large after random forests and then calculated. At the same time, the delayed attacks of malicious nodes are ignored (*Sun, Ruan & Liu, 2019*). By analyzing the creation of contracts, invocation of contracts, and Ethereum transactions in Ethereum transactions, it is found that malicious nodes create spam contracts in large quantities (with similar contract content and few invocations), and by clustering known malicious nodes with a vector-space distance based on vector space distances and analyzing the potentially malicious nodes out, but the dynamic timeliness of the nodes is missing the observation of the dynamic timeliness of the nodes. *Sajid et al. (2022)* uses blockchain to record node registration information and transaction information and proposes a malicious node detection model based on Genetic Support Vector Machines and Genetic-Based Decision Trees, which is capable of detecting the grey hole attack abuse attack and MITM attack. An interval-based blockchain model is introduced, which uses network block monitoring nodes to analyze the blocks generated by nodes at specific time intervals. This scheme satisfies the dynamic timeliness of malicious node detection but does not detect malicious behavior at the level of node response to events. *She et al. (2019)* implements blockchain smart contracts and wireless sensor network quadrilateral measurements for malicious node localization detection and records the consensus results

of the voting in a distributed blockchain, which ensures the traceability of the detection process but does not detect the malicious nodes in the blockchain. *Farrugia, Ellul & Azzopardi (2020)* among malicious and normal nodes labeled by the Ethereum community, using the XGBoost classifier to detect illicit accounts based on transaction history, and finally, with 10-fold cross-validation, XGBoost achieves an average accuracy of 0.963. This method can distinguish illegal users in Ethernet well; the method is applicable to detect malicious nodes with labels, and no detection is done for malicious nodes without labels.

From the above analysis of related studies, it is concluded that there are the following shortcomings for malicious node detection for SIoT combined with blockchain:

1) Comparison of malicious and normal nodes is not involved in the detection system;
2) The detection system does not address the subjectivity, dynamic timeliness, and privacy protection of the judgment nodes;
3) Lack of malicious behavior detection for SIoT users.

In this article, subsequent experiments will address each of these issues.

## SIoT and blockchain technologies and principles

This section focuses on two background technologies utilized in this article, which will be described in detail here.

### SIoT overview

SIoT is a network of smart things identified by unique identifiers (UIDs) (*Aftab et al., 2020*) and connected that can communicate without human interaction.SIoT is an emerging concept that converges social networking and IoT, aiming to provide rich, smart, and socially-enabled service experiences by integrating social interactions with IoT technologies (*Shamszaman & Ali, 2017*). Its application areas have expanded from standard areas to large-scale industries such as smart manufacturing, supply chains, smart grids, food industry, healthcare, and connected cars. Thus, SIoT is seen as a key element in shaping the future smart, connected world.

SIoT system is a network of smart things identified by unique identifiers that can communicate without human interaction. SIoT security refers to the protection of SIoT devices and the various devices that support connectivity to the Internet, as well as the protection of data generated by SIoT systems, privacy, and many other aspects. SIoT security is a continuously evolving field, and protecting the security and reliability of SIoT systems becomes even more critical as technologies evolve and threats change. A holistic approach to security measures that considers physical, cyber, and data security is essential to ensuring the security of SIoT systems. In this article, we combine SIoT with the consortium blockchain and then use machine learning to analyze the malicious nodes and lay the foundation for future trust management by identifying the malicious nodes, thus ensuring the security of SIoT users as well as the integrity, confidentiality, non-repudiation, traceability, and availability of the data.

## Overview of blockchain technology

Blockchain technology initially originated from Bitcoin and is the core support technology for virtual digital currencies such as Bitcoin, aiming to solve the problem of how to build a "trust" ecosystem to meet the needs of activity occurrence and development in the absence of a credible centralized institution, as well as information asymmetry and uncertainty (*Dong et al., 2023*). Later, Vitalik Buterin pioneered Ether, and blockchain technology came to the smart contracts age, making the blockchain programmable and leading to the birth of decentralized applications (*Buterin, 2014*). Later, the open-source distributed ledger platform Superledger, directed and created by the Linux Foundation, opened the era of the blockchain's consortium blockchain, allowing the blockchain to expand and extend to more areas (*Pilkington, 2016*; *Dong et al., 2023*). The underlying principle is a distributed ledger technology that forms a chain of continuously lengthening blocks of data by linking transaction records together in a kind of sequence, containing in each block the hash value of the previous block, forming a tamper-proof blockchain structure at a time. The blockchain is equivalent to a distributed database in every participant or trader's computer node, and every transaction will be recorded on the blockchain (*Abdelghani et al., 2016*). This blockchain structure ensures data security, transparency, and comparability.

In this article, the consensus algorithm used by the consortium blockchain is the Proof of Authority (POA) algorithm, which uses a specific authenticator to validate and create new blocks rather than relying on computational power or token equity. POA algorithm can provide high performance and low energy consumption for SIoT systems. POA does not require many computational resources, and it can achieve efficient transaction processing speed and low energy consumption, which also realizes the lightweight nature of SIoT. Consortium blockchain can also bring more effective security protection mechanisms for SIoT's data, giving SIoT safe and reliable information storage and management means. When the consortium blockchain is applied to SIoT, it should consider the node's private data protection, lightweight security mechanism, malicious node detection and defense, automated access control policy, and system scalability. In this article, we focus on detecting malicious nodes and data privacy protection, and the specific detection method is to distinguish malicious nodes from normal nodes to provide a secure foundation for future work.

The security and efficiency of SIoT systems and the reduction of their costs can be significantly enhanced by applying a consortium blockchain to them. The advantages of consortium blockchain in SIoT are explored in this article.

1) Privacy protection: Consortium blockchain can ensure that sensitive data is only visible to authorized participants. This is important for situations involving personal privacy in SIoT;

2) Transaction transparency: consortium blockchain provides a transparent record of transactions, with all participants able to view the history of transactions, and this transparency can help build trust and be more traceable;

3) Consensus mechanisms: The POA consensus mechanism can ensure data consistency and credibility and prevent malicious nodes from damaging the system. This mechanism can provide high performance and improve system efficiency;

4) Smart contract enforcementfast trading and settlement: Consensus mechanisms and optimized design enable fast transaction confirmation and settlement, accelerating interaction and business processing between SIoT devices;

5) Data sharing and collaboration: The consortium blockchain can provide a secure data sharing and collaboration mechanism to promote cooperation between SIoT devices, such as joint monitoring and automated collaborative operation;

6) Reduced transaction costs: Automating processes and reducing intermediaries through smart contracts can reduce the cost of transactions and data processing and increase efficiency.

## DEFINITION OF MALICIOUS NODES AND INTRODUCTION TO RELATED ALGORITHMS

A malicious node is a device or node with negative behavior or intent in SIoT. Because of the large number of Internet devices and sensors designed in the SIoT, malicious nodes pose a severe threat to the security and stability of SIoT systems. Malicious nodes in the SIoT system will create attacks such as data tampering, negative transmission, denial of service, attacks on other nodes, malicious access to data, and other attacks that are not conducive to the development of the SIoT system.

In real life, malicious nodes can carry out many attacks in social IoT, such as tampering with data, denial of service, negative transmissions, and malicious access to data. However, our consortium blockchain system is an environment where data is open, transparent, and tamper-proof. This curbs many malicious attack issues, such as data tampering and malicious access to data. For example, attacks such as denial of service, harmful transmission, *etc.*, essentially reduce the processing efficiency of the system, which our proposed transaction delays and transaction intervals can detect.

### Definition of malicious node

In this section, we are going to define the malicious nodes.

(Definition) Malicious nodes: In the consortium blockchain based SIoT system, we will determine whether a node is a malicious node or not by a combination of delayed or negative services and categorize such nodes as malicious nodes.

(Definition) Delay service: Delayed service implies that there is an intentional delay in service or delay in transaction by the node. In specific data, the presence of delayed service in that node can be detected by our proposed algorithm.

(Definition) Interval: The distance between any behavioral record of a node and the most recent behavioral record of a node within the selected block interval is taken as the interval of the node.

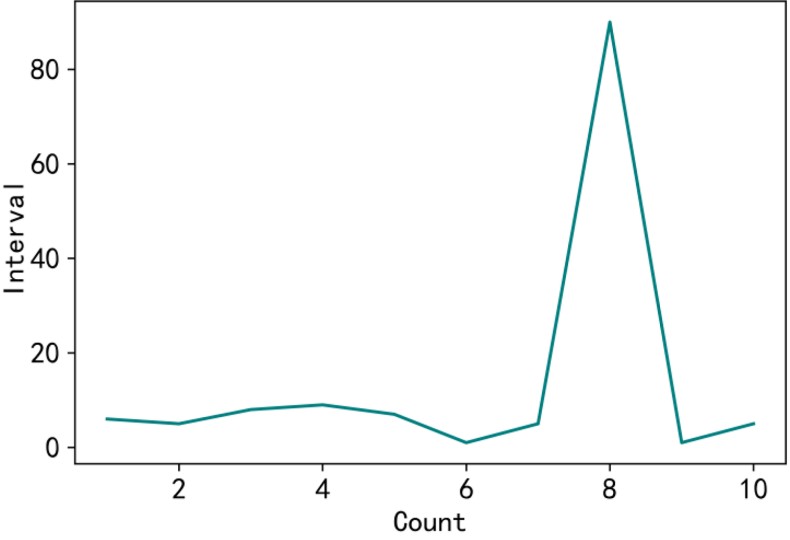

**Figure 2  Test node.**

(Definition) Negative service: Negative service exists for a node if there is an anomaly in the transaction interval of the node over a period of time.

In the consortium blockchain network, the operation record and operation time of each user can be obtained, in which each block can record the operation record of some users, and the record time has a sequential order. If a node has negative service in the specified block interval, the node may have malicious behavior in this operation. Anomalous intervals are reflected in excessive fluctuations in node operation intervals, as shown in Fig. 2. When the inter-block interval is significantly large between the eighth interaction, it can be deduced that the node produced a negative service during that period.

## Handling delay algorithm

This chapter will specify the proposed node processing delay algorithm. It mainly reflects whether the interaction interval of distributed SIoT nodes over a period of time is abnormal or whether delayed services occur. In the consortium chain, the block number of each block is sorted in chronological order. In addition, each block has a corresponding hash address. The associated processing delay of that node can be calculated by obtaining the block number and hash address of each block and the operation records of each node.

Suppose A is an SIoT user on the consortium blockchain. In the selected time domain T (t), the collection of all operation records for A is X:

$$X = \{x_1, x_2, \ldots, x_n\}.$$

The operation interval can be expressed as:

$$Y = \{(x_2 - x_1), (x_3 - x_2), \ldots, (x_n - x_{n-1})\} = \{y_1, y_2, \ldots, y_{n-1}\}$$
$$Y_{\text{avg}} = \frac{1}{n-2} \sum_{n=1}^{n-2} Y.$$

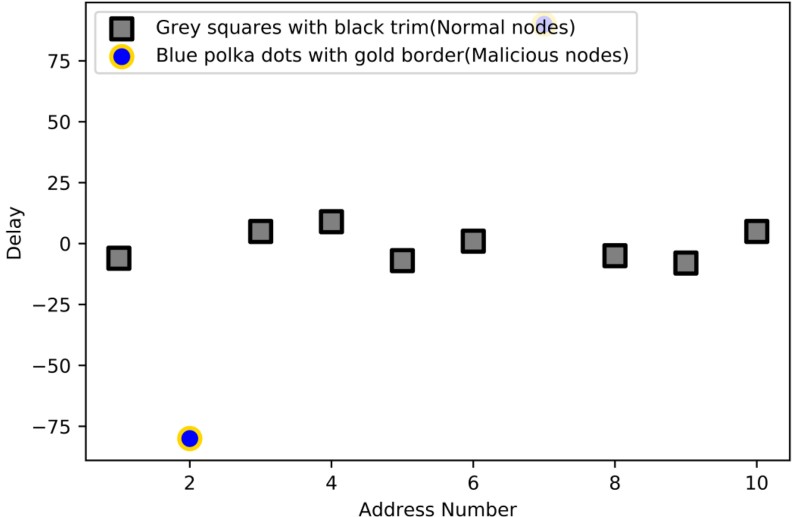

**Figure 3  Differentiate nodes.**

So, the processing delay formula is as follows:

$$delay = \ln(|Y_{\mathrm{avg}} - y_{n-1}| + 1) \cdot \mathrm{sign}(Y_{\mathrm{avg}} - y_{n-1})$$

where sign(x) is the sign function, which returns the sign of the argument x, *i.e.*:

$$\mathrm{sign}(x) = \begin{cases} -1 & \text{if } x < 0 \\ 0 & \text{if } x = 0 \\ 1 & \text{if } x > 0 \end{cases}.$$

Delay is a mathematical calculation of the difference between a node's transaction intervals and its historical average transaction intervals for all transactions. Our latency algorithm determines whether a node is anomalous by comparing the node itself with its history. A node is not directly determined to be malicious when its computational resources are low but rather when the node makes its historical average transaction interval change abnormally in a single transaction.

The evaluation criteria for the result are as follows: When the delay is greater than 0, the average interval in the time domain, except the last interval, is greater than the previous interval, indicating that the current service duration is longer than the average. Currently, the corresponding node A can be judged as the present positive; otherwise, node A is the recent negative. When the delay is much greater than 0, it can be seen that the last operation interval is positive. Still, if an operation before this period is negative, it is determined that the node is negative in this period. When the delay is much less than 0, the last operation interval is negative, but the previous operation interval is positive. Considering that distributed network nodes have some network delays, the delay of normal nodes should fluctuate around zero. Figure 3 shows the distribution of malicious and normal nodes with different delay values.

## EXPERIMENTS AND RESULTS ANALYSIS

In SIoT, the behavior of nodes is dynamic, and they may exhibit different behavioral traits at different times, including the transition from normal to malicious nodes. This dynamism and uncertainty are challenging for methods to detect and predict. Therefore, our dataset intercepts the transaction data of the consortium blockchain for one time period to detect whether a node is abnormal in that period. A node's behavior may change over time. For example, a node is detected as a normal node in a recent period but may become a malicious node in another period. This situation exemplifies the uncertainty and dynamic timeliness of our method's detection. Our method can detect abnormal nodes in one time period, but due to the dynamic nature of node behavior, we cannot guarantee that the node will not change in the subsequent period. Therefore, we need to regularly monitor and evaluate the behavior of nodes to detect and respond to possible anomalies in a timely manner to guarantee the system's security and stability. In this regard, machine learning techniques are essential in determining the relationship or malignancy between individual nodes based on information such as the roles, behaviors, and needs of the participants or nodes. In this article, by combining the SIoT system through the consortium blockchain, the address represents the account, the transaction recorded by the address posting to the consortium blockchain represents the operation, and the records generated by the same address on different blocks represent the operation intervals. In many block information and node operations, unsupervised algorithms in machine learning perform clustering to distinguish malicious nodes from normal nodes, and the clustering results classify nodes into malicious nodes and normal nodes.

### Data set introduction and preprocessing

In SIoT, user transactions and services play a crucial role. Transactions and services are regarded as essential carriers of economic activities and user value creation in SIoT, driving the development and prosperity of the entire SIoT ecosystem. By detecting transaction and service delays, user experience and system efficiency can be improved, and the stability and reliability of the SIoT system can be enhanced. Excessive service or transaction latency may lead to resource wastage and cost increase. For example, system resources are still occupied while the user is waiting, but no effective service can be provided, leading to wasted resources and increased costs. At the same time, protecting the privacy of SIoT users is fundamental to building trust and ensuring security. Our dataset focuses on transactions and services executed by SIoT nodes and captures only transparent and tamper-proof transaction data on the consortium blockchain. This not only ensures user privacy but also the trustworthiness and accuracy of transactions.

To stay close to the transaction or service process in SIoT, we build a consortium blockchain based on Ethereum technology, adopting PoA as the consensus algorithm. PoA is a highly efficient consensus mechanism that verifies the transaction and packages the blocks through authorized nodes, ensuring the security and high performance of the network. On our consortium blockchain, the nodes participating in the consensus are licensed, and this consensus mechanism can be better adapted to the needs of our specific scenarios while guaranteeing the decentralization and security of the consortium

| | address | block_number |
|---|---|---|
| **0** | 0x0000000048429ba5463a4a9aa866460087dcebd0 | 29026 355.0 29395 351.0 63056 22.0 63301 20.0 Name: block_number, dtype: float64 |
| **1** | 0x00000000863b56a3c1f0f1be8bc4f8b7bd78f57a | 3738 601.0 4719 592.0 47466 180.0 47491 180.0 47494 180.0 47531 180.0 Name: block_number, dtype: float64 |
| **2** | 0x000000062bf07241987d5f1f88fc25c9190bcfc9 | 9894 539.0 39350 258.0 Name: block_number, dtype: float64 |
| **3** | 0x00000006e42915a2b6907f8b3faf311b68862f60 | 51474 140.0 58136 73.0 Name: block_number, dtype: float64 |
| **4** | 0x000000697bb288f2528042e8844b65cd32beafca | 7524 564.0 8501 554.0 34298 303.0 Name: block_number, dtype: float64 |
| **5** | 0x000000b833f5d0d1119c479df0777789a6ac207d | 34352 303.0 36379 285.0 37055 278.0 Name: block_number, dtype: float64 |
| **6** | 0x0000098a312e1244f313f83cac319603a97f4582 | 23766 403.0 52575 129.0 Name: block_number, dtype: float64 |
| **7** | 0x000101d1c9769ea22d6ea627b19bffcc99f61bf3 | 52437 130.0 53892 117.0 54086 115.0 Name: block_number, dtype: float64 |
| **8** | 0x0001276ab06569d7a6582ee5367345809522adbc | 52763 127.0 55750 97.0 Name: block_number, dtype: float64 |
| **9** | 0x0002b5ca4da0bf9b22480a39c32145159388baed | 2378 613.0 3442 603.0 4621 593.0 8773 551.0 10096 537.0 36529 284.0 Name: block_number, dtype: float64 |

**Figure 4 Transaction information.**     

blockchain. We generate and record data by executing transactions and smart contracts on the consortium blockchain and then intercepting data from the consortium blockchain over time. We will compare the transaction data on the consortium blockchain to the transaction data between SIoT users, where each address account corresponds to the resource user in the SIoT system, and each transaction information recorded in the consortium blockchain is the information of the SIoT user's transaction resources.

Nodes are required to obtain authoritative permission to join the consortium blockchain, which reduces the likelihood of malicious nodes joining the system. Our dataset is derived from real transaction data in Ethereum, an approach that ensures the authenticity and diversity of the data. As an open and transparent blockchain network, Ether's transaction data has a high degree of authenticity and accurately reflects the actual transaction behaviour and patterns on the blockchain. Moreover, the dataset on Ether is very rich, representing all possible malicious nodes in the real world. It also ensures the immutability and authenticity of malicious node behaviour. Modelling this data onto the consortium blockchain allows you to make the dataset more representative, covering different types of transactions and usage, thus increasing the diversity and value of the data. The period data has a total of 634 blocks recording 100,000 transactions. Firstly, empty entries in the data are cleaned, valid from_address and block_number fields are extracted, and each address in from_address is used as an indexing condition. The corresponding block/number is added to the corresponding address. As shown in Fig. 4, we can get the transaction information of all operations related to each account address during this period.

At this time, the amount of data is compressed to 12,418 rows. After removing the rows with a number of transactions less than 2 in the period, there are 5,899 rows left, which

| | address | interval | delay |
|---|---|---|---|
| 0 | 0x0000000048429ba5463a4a9aa866460087dcebd0 | 331.000000 | 161.500000 |
| 1 | 0x00000000863b56a3c1f0f1be8bc4f8b7bd78f57a | 412.000000 | 94.000000 |
| 4 | 0x000000697bb288f2528042e8844b65cd32beafca | 251.000000 | 241.000000 |
| 5 | 0x000000b833f5d0d1119c479df0777789a6ac207d | 7.000000 | -11.000000 |
| 7 | 0x000101d1c9769ea22d6ea627b19bffcc99f61bf3 | 2.000000 | -11.000000 |
| 9 | 0x0002b5ca4da0bf9b22480a39c32145159388baed | 319.000000 | 69.750000 |
| 11 | 0x000f422887ea7d370ff31173fd3b46c8f66a5b1c | 574.000000 | 8.133333 |
| 12 | 0x0010985214e2a95244b4765b7df8693d1c5c393f | 7.000000 | 2.000000 |
| 14 | 0x001c356c0be5dd6c91ca24ef04d9e10081510682 | 40.000000 | -15.000000 |
| 16 | 0x001cb8d33538dbba078e5644735bc1ece4004dd1 | 25.000000 | 10.500000 |
| 18 | 0x0023af05db6f207a76129e21e291401f8d007f40 | 8.000000 | -66.000000 |
| 21 | 0x0032857aec7f3833108362ad780c73979518a4c5 | 28.000000 | -50.000000 |
| 23 | 0x0037825fd75af7eeace28889665e3fac8fdb6300 | 118.000000 | -146.000000 |
| 26 | 0x0039b625b1d8632c7a0057c964ec58a9f39789d3 | 389.000000 | -34.166667 |

**Figure 5  Transaction information.**

reduces the occasionality of the experimental results. According to the processing delay algorithm mentioned in the previous chapter, the acquired block data is calculated using the algorithm, and finally, the delay value and interval value are obtained. As shown in Fig. 5.

The scatter plot is shown in Fig. 6; the horizontal axis represents the block processing interval of the node, and the vertical axis represents the delay value obtained by the node through the algorithm.

## Specific experiment and analysis

In SIoT, an encounter history-based discovery approach is mainly used: objects are discovered by exploiting the long-term social relationships between them (*i.e.*, things that have been frequently encountered or encountered in the past). Therefore, the objects in SIoT are to be analyzed through historical data, and the malicious nature of the nodes obtained in the previous section is used to distinguish between malicious and normal nodes. Firstly, the K-means algorithm and Density-Based Spatial Clustering of Applications with Noise (DBSCAN) algorithm in the unsupervised learning clustering algorithm are used to cluster the data. Then, the most reasonable algorithmic scheme is selected by comparing the resultant effects of the two algorithms.

K-means clustering: The algorithm divides the data set into K distinct clusters, each representing a data set. The algorithm's goal is to minimize the sum of the square distance

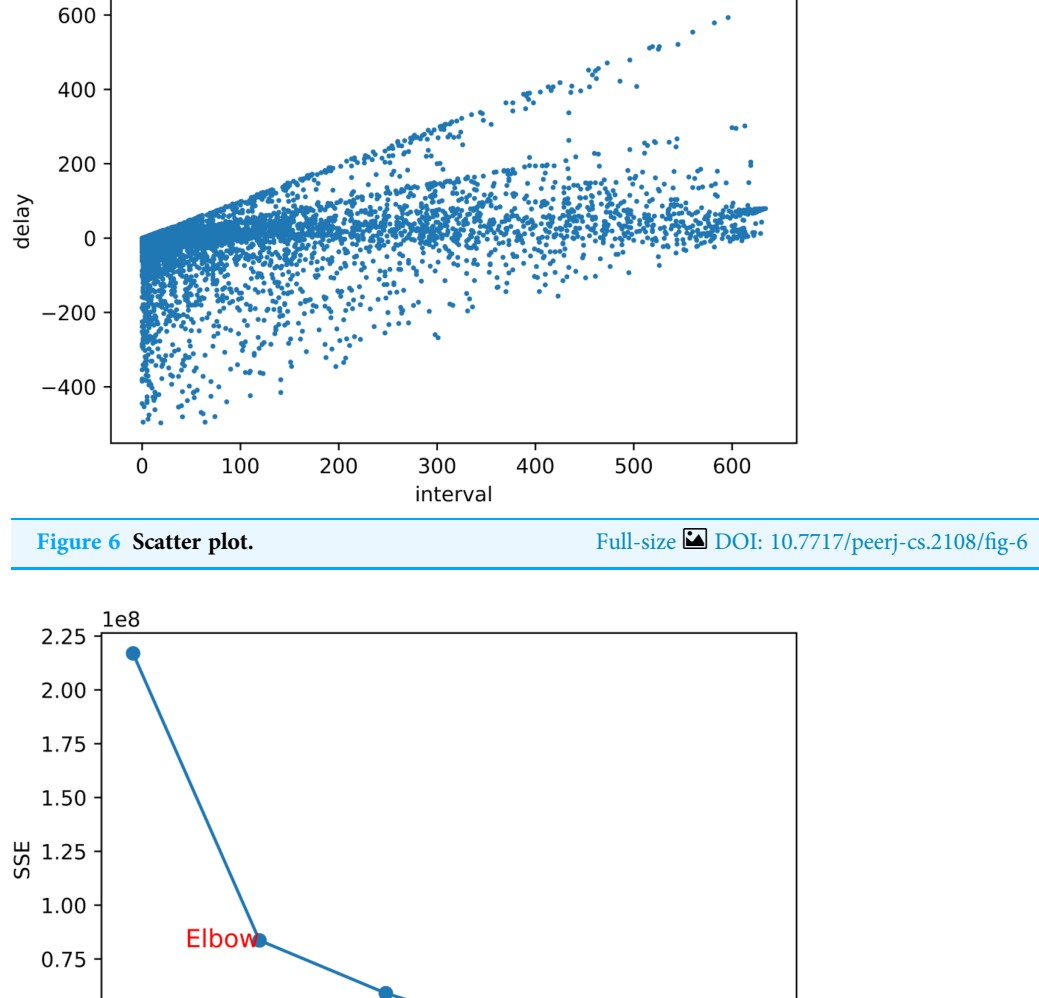

**Figure 6  Scatter plot.**

**Figure 7  Elbow method.**

between the data and the center point of the cluster to which it belongs. Therefore, the clustering method must set the k value to distinguish the malicious node cluster from the normal node cluster. Considering that different k values can produce other clustering effects, we use the Elbow Method, Silhouette Coefficient Score, and Calinski-Harbasz Score to compare and analyze the best k value.

Elbow method: The elbow method selects the optimal K value by drawing the relationship between the number of clusters K and the corresponding sum of squares within the cluster, usually choosing the K value at the inflection point as the optimal number of clusters. As shown in Fig. 7, its inflection point should be at k = 2, and then the value of k is selected as 2.

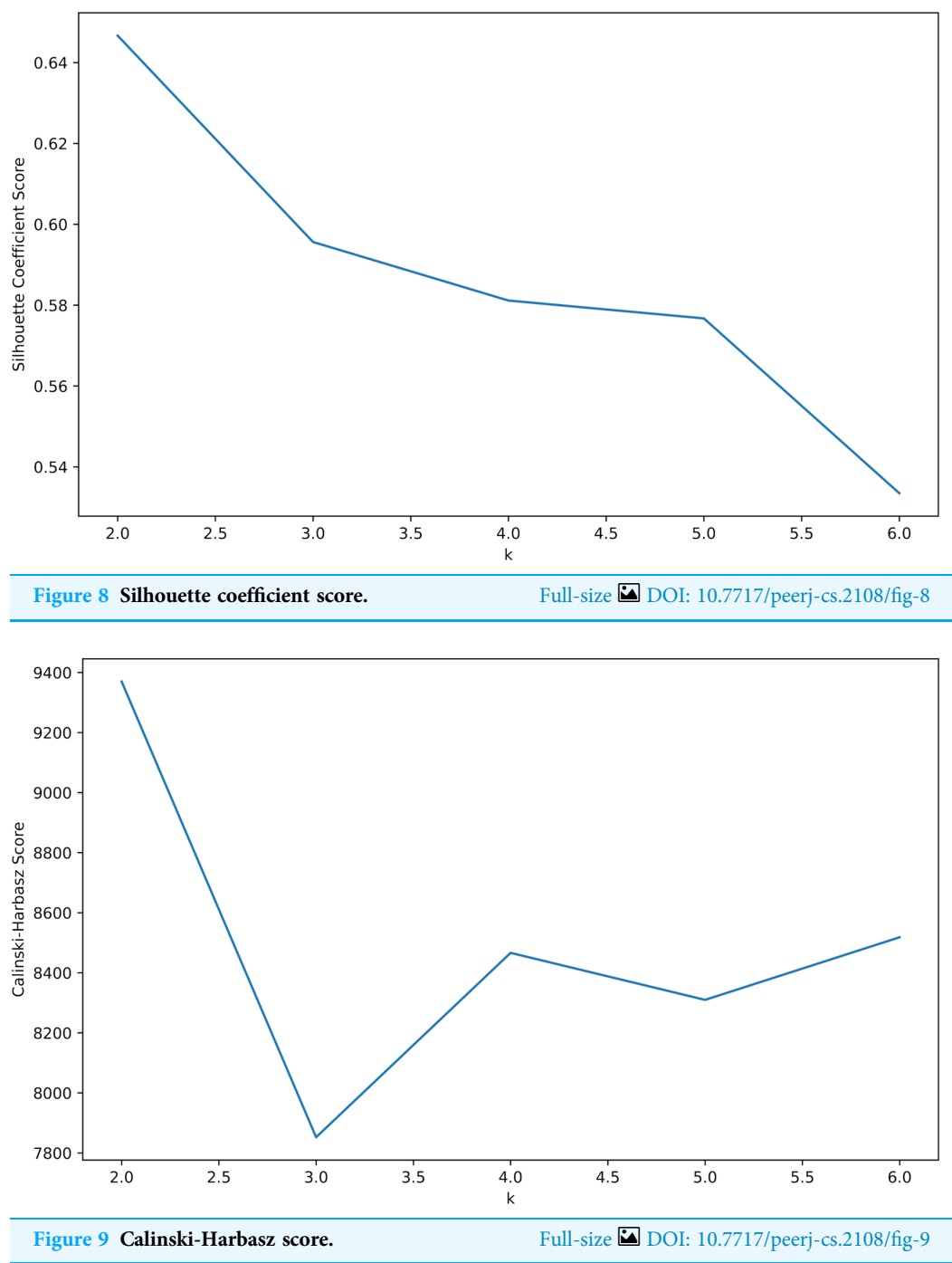

**Figure 8  Silhouette coefficient score.**     

**Figure 9  Calinski-Harbasz score.**     

Silhouette coefficient score: The Silhouette coefficient score measures cluster density and separation in the range of [−1, 1]. The larger the value is, the better the cluster distribution is. As shown in the Fig. 8, when k = 2, its score is the highest, then the k value is chosen as 2.

Calinski-Harbasz score: Calinski-Harabasz score is an index used to evaluate cluster quality. The higher the score, the more compact and dispersed the clusters are. As shown in the Fig. 9, when k = 2, its score is the highest, and the k value is chosen as 2.
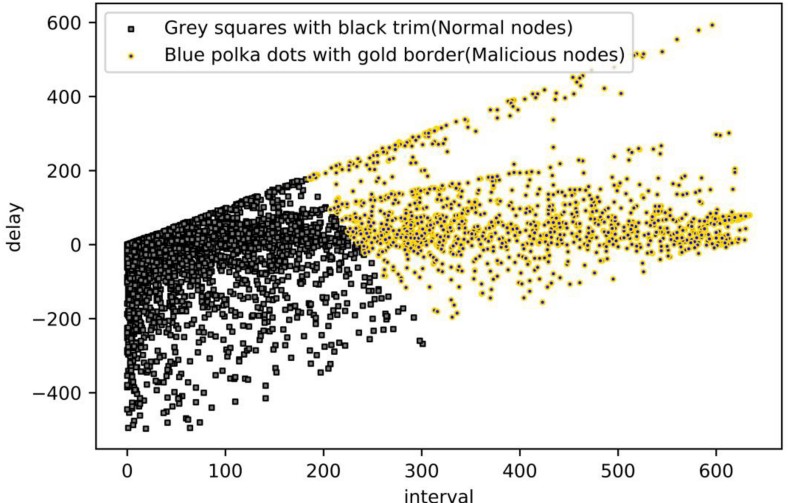

**Figure 10  Malicious nodes detection from K-means.**

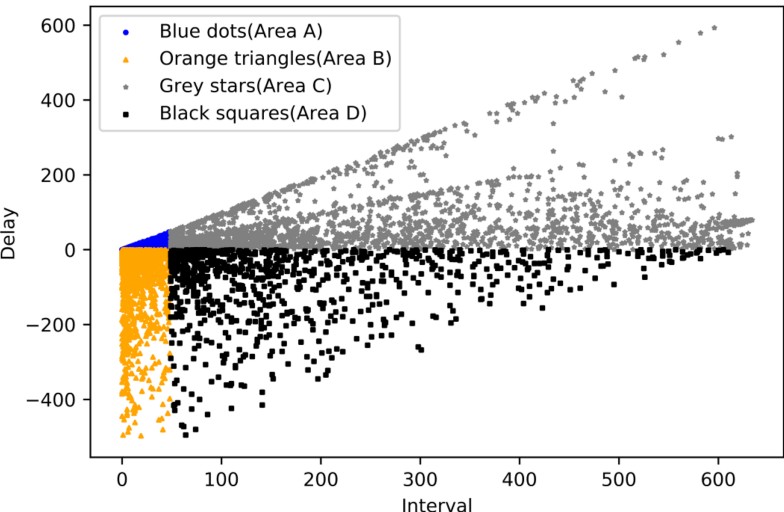

**Figure 11  Malicious nodes area.**

To summarize, k-means clustering works best when the value of k is assigned to 2, so the value of k is assigned to 2. Its clustering result graph is shown in the following Fig. 10: Blue polka dots with gold border markers are normal nodes, and grey squares with black trim markers are malicious nodes.

DBSCAN clustering: DBSCAN is a density-based clustering algorithm that identifies regions with sufficiently high density and treats these regions as clusters. DBSCAN does not require a predetermined number of clusters. It can find clusters of arbitrary shape and recognize noisy points (points that do not belong to any cluster). The clustering algorithm has two parameters centered on the initial matter, the radius of the field of that point R and the minimum number of points in the area MinPoints. Since the method is a density-based unsupervised clustering method, no specific evaluation metric exists. However, in the previous chapter, the evaluation criteria for processing delay algorithms were analyzed, *i.e.*,

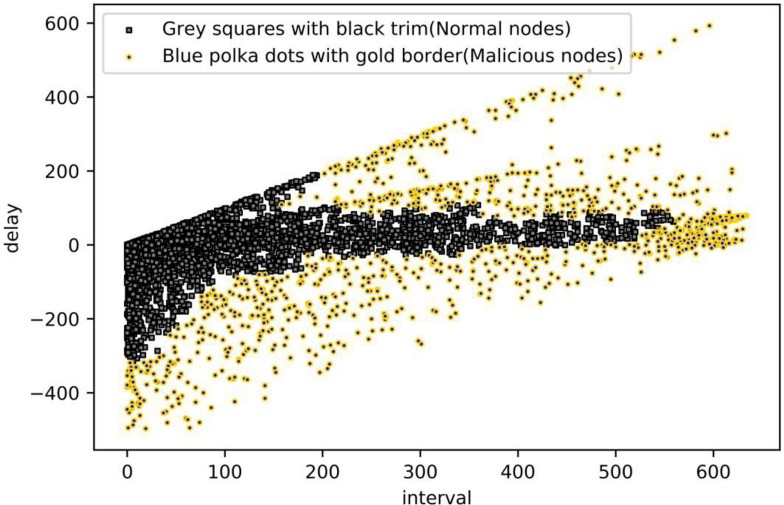

**Figure 12  Malicious nodes detection from DBSCAN.**

nodes with delay fluctuating around 0 are normal nodes. In contrast, the evaluation criteria for interval is judged by the block intervals, where the smaller the interval, the more positive the node is, and *vice versa*, the more negative the node is. To better realize the clustering, The data scatter plot is divided into four parts, with blue dot nodes representing services that are faster than average and have shorter-than-average service intervals, grey star areas representing services that are faster than average but have longer-than-average service intervals, orange triangle nodes representing services that are slower than average but have shorter-than-average service intervals, and Black square nodes representing services that are slower than average but have shorter-than-average service intervals. As shown in Fig. 11.

Next, each of the four areas is clustered according to the DBSCAN algorithm steps through the optimization of the parameters, in which Area A should be all normal nodes; R in Area B is set to 15, and MinPoints is set to 15; R in Area C is set to 10, and MinPoints is set to 20; R in Area D is set to 10, and MinPoints set to 15. The final result is obtained by clustering, where the blue polka with black trim means that the node is judged as a malicious node in the current time period, and the grey squares with black trim is the opposite. As shown in Fig. 12.

Based on the analysis of the results obtained by the two clustering algorithms, the density-based clustering algorithm DBSCAN algorithm can classify the processing delay as the evaluation criterion. In contrast, the distance-based clustering k-means algorithm can obtain the best classification effect under binary classification. Still, the generated results do not consider the evaluation criterion of processing delay. Therefore, considering the processing delay algorithm and interval, the clustering effect of the DBSCAN algorithm is the most satisfactory.

The final clustering result will get a data set, one of which is a malicious node and one of a normal node. The two data sets are set to Boolean type format, where True indicates normal node and False indicates malicious node. It also includes the original transaction

---

**Algorithm 1    Uplinking process.**

**Require:** *OAddress* (Original address), *AAddress* (On-blockchain address), *BValue* (Boolean value)

Create mapping variable *addressMapping* to store the mapping from the original address to the on-blockchain address.

Create mapping variable *booleanMapping* to store the mapping from the original address to the Boolean value.

**while** Data set is not empty **do**

    Retrieve *OAddress* and *BValue* from the dataset

    Find *AAddress* by calling the lookup function with *OAddress*

    **if** *AAddress* is not empty **then**

        Store the mapping from *OAddress* to *AAddress* in *addressMapping*

        Store the mapping from *OAddress* to *BValue* in *booleanMapping*

        **if** *OAddress* == *AAddress* **then**

            Link the Boolean value to the corresponding *AAddress*    Perform the linking operation

        **end if**

    **end if**

**end while**

---

address and the corresponding consortium blockchain address. Data upload: The data is uploaded to the consortium blockchain through the smart contract function, receives the original transaction address, and stores it in the smart contract. And write a query function or interface, allowing users to query the corresponding consortium blockchain address through the original transaction address. As shown in Algorithm 1.

We take the data obtained from clustering and uplink it through smart contracts. In this algorithm, OAddress represents the original address, AAddress represents the on-blockchain address, and Bvalue represents the Boolean value of the two kinds of data. The specific process of the smart contract algorithm is:

1) Create mapping variable address mapping to store the mapping from the original address to the consortium blockchain address;

2) Create the mapping variable booleanMapping to store the mapping from the original address to the Boolean value.;

3) Firstly, it detects whether the data in the dataset is empty; if it is empty, it means that the data up-chaining is finished. When it is not empty, store the mapping from the original address to the address on the blockchain in addressMapping and store the boolean value of the original address in booleanMapping. Read the corresponding consortium blockchain address from address mapping based on the original address. Read the boolean value from booleanMapping according to the original address, then check whether the original address and the address on the blockchain are the same; if they are the same, then the boolean value will be uplinked to the corresponding address of the consortium blockchain, and repeat this operation until the dataset is empty;

After we create the mapping, the time complexity of its lookup is O(1), significantly improving the lookup efficiency. Also, storing the data in the blockchain provides the following advantages.

1) Sharing data and cooperation: Allowing multiple organizations to co-manage and share data facilitates cross-organizational collaboration, where parties can view, verify, and participate in transactions on the blockchain. On the blockchain, not only can trust be built, but also the cost of cooperation and efficiency can be reduced;

2) Data traceability and non-tampering: Blockchain technology ensures the immutability of data, which cannot be altered once uploaded. This helps ensure trust in the data and reduces the risk of fraud and data tampering;

3) Privacy and security: Consortium blockchain offers a higher level of privacy and security compared to public blockchain, where only authorized nodes can process transactions on the blockchain;

4) Smart contract enforcement: Smart contracts on the consortium blockchain can automate business logic and perform operations based on predefined conditions. This helps to reduce human error, increase efficiency, and reduce operational costs.

In this article, the data of the malicious node will be recorded on the consortium blockchain after it is uploaded. Thus, it will be tamper-proof and provide a transparent history. This is crucial for detecting malicious nodes and behaviors and avoiding tampering or falsifying data. At the same time, the consortium chain offers a platform for multiple participants to share trusted data. SIoT users can share data and work together to detect malicious nodes, creating a win-win situation for cooperation and facilitating users to judge whether to conduct a transaction at a node by checking whether the node has any negative behaviors and, thus, whether to complete the transaction at that node. Finally, uplinking ensures auditable and traceable data and provides a reliable way for other users to discover better service providers.

## Comparative analysis of experiments

We use real malicious node data for detection and comparison and form 100 consortium blockchain nodes through laboratory equipment, through which transactions or services are executed. We designed 20 true malicious nodes out of 100, mainly through delayed attacks, transaction interval attacks, denial of service attacks, negative transport attacks, and hybrid attacks. These nodes form a history through transactions on the consortium blockchain, which contains thousands of transaction histories of the nodes, and we detect the nodes through their history. We have collected a large amount of data, amounting to thousands, on each node. This data contains information about the communication, interaction and behaviour between nodes and is key to our research. Our goal is to use this data to validate the effectiveness of our proposed approach in defending against specific types of attacks, and to ensure that our experimental results are reliable and reproducible. By adequately collecting and analysing a large amount of data, we can gain a

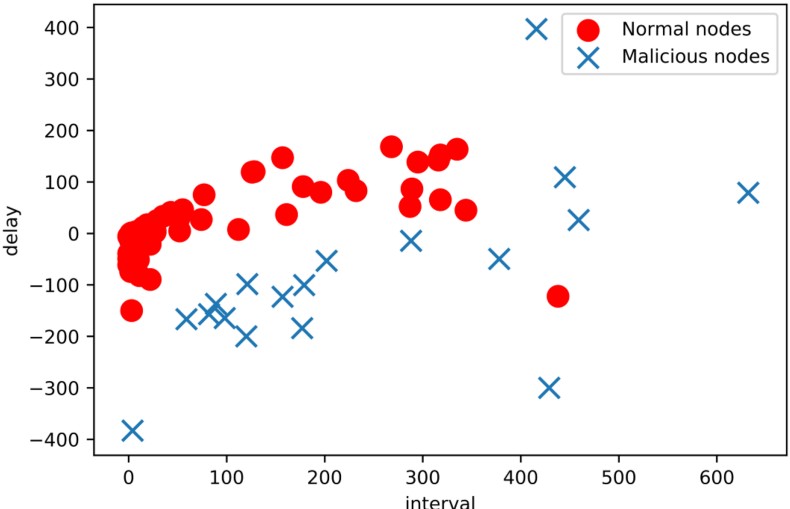

**Figure 13 Malicious nodes detection from DBSCAN.**

comprehensive understanding of the effectiveness of our approach (*Frustaci et al., 2017*). The main attack modes are as follows:

1) Delay attack (*Wei, Yuan & Zheng, 2018*): Nodes deliberately send transactions or messages with different delays to the network;

2) Transaction interval attack (*Narang & Kar, 2021*): Nodes interfere with the network by inserting abnormal time intervals in transactions;

3) Denial of service attacks (*Lau et al., 2000*): In a denial of service attack, a malicious node attempts to disrupt normal network communication by sending a large number of invalid transaction requests;

4) Negative transmission attack (*Yin et al., 2022*): In a negative transmission attack, a malicious node intentionally discards valid data transmitted by other nodes, thereby interfering with network communication or causing data loss;

5) Hybrid attacks (*Liu et al., 2021*): Multiple attacks occur simultaneously, such as sending a large number of invalid requests while dropping valid transaction data transmitted by other nodes.

Using a combination of these methods, a wide range of malicious behaviours can be simulated, which helps in evaluating the robustness and efficacy of malicious node detection methods. We have successfully detected the simulated malicious nodes using the proposed model. As shown in Fig. 13.

During our detection process, we set the tag value of the malicious node to 1 in advance and at the same time, we set the tag value of the node that is detected as malicious to −1. As shown in the Table 1, we will compare the real malicious node with the detected malicious nodes shown in the table. Where Est equal to −1 means detected malicious node and Label equal to 1 means real malicious node. We can clearly see that 18 nodes are detected as malicious nodes, and the remaining 82 nodes are normal nodes, as shown in Table 2. We

**Table 1 Comparison table of malicious nodes**

| Address | Interval | Delay | Est | Label |
|---|---|---|---|---|
| 0x01c48f1fa4061301aa19b7f09b9f6a8756bf49d9 | 4 | −383 | −1 | 1 |
| 0x01eb62ff431b4782e34b76cc801d4c385c4049ee | 445 | 109.25 | −1 | 1 |
| 0x021d9be5e82353a2b609b0842e5e58ee2af05f32 | 416 | 397 | −1 | 1 |
| 0x06e14b037650e6d7c5e6a8eb329e1b44e2d7650c | 632 | 79 | −1 | 1 |
| 0x0711cddb105819b14d55439581fe092aa5b39461 | 459 | 26 | −1 | 1 |
| 0xdd05669c524b2633d76b446fb54aa756212563d8 | 177 | −184 | −1 | 1 |
| 0xdd2bc15318d884094e9b3b29733bff42c964b577 | 378 | −49.64 | −1 | 1 |
| 0xdd498f56dde61e47e69ee9b2561b462baf48d9b9 | 157 | −123 | −1 | 1 |
| 0x01de494230b03989f9a2d8956fdfe5486d712ebf | 121 | −98.5 | −1 | 1 |
| 0x01f0831120ab81f91109e099afb551a091c4c05a | 82 | −157 | −1 | 1 |
| 0x020717ec52380aa0549cf7bf9f5976274371a58a | 89 | −136.75 | −1 | 1 |
| 0x0215e80a798723dba358205a0765732643c7d34a | 59 | −166.5 | −1 | 1 |
| 0x06f8d80e285d00b7c65db2a999a290d9636e9c6f | 98 | −164.4 | −1 | 1 |
| 0x0710ee20cb035b50f6aedf58bf37b8dbd842f1ed | 120 | −200 | −1 | 1 |
| 0x0717ba2152b8b4763e3ea1fdea7eebf16c54070b | 429 | −300 | −1 | 1 |
| 0x074dccdb202ac9fe0f4b25bdaa33762287579ea2 | 288 | −14.43 | −1 | 1 |
| 0x0758e77def8db3e078dd579a2e686a4543e5d6eb | 179 | −100.25 | −1 | 1 |
| 0x0971fcb069ecde4eb4a43325e46041aab73f5732 | 202 | −53.33 | −1 | 1 |

**Table 2 Comparison table of malicious nodes.** Total 82 rows.

| Address | Interval | Delay | Est | Label |
|---|---|---|---|---|
| 0x0973ff3cbf6b229380f702f7e2af44d377e6be96 | 3 | −28 | 0 | 0 |
| 0x0998eeec2a32c6160c962a9338627263a6f35341 | 9 | −4 | 0 | 0 |
| 0x09a0da7289a60809992d819530833f4053124090 | 3 | −150 | 0 | 0 |
| 0x09aa7bd101451c9e1d99764dbb4846cb08cdbc02 | 22 | −89.5 | 0 | 0 |
| 0x09d486ed40727a45fd394d2e658d88a0990b2455 | 4 | −2 | 0 | 0 |
| 0x0a0fc35cc2c544de676bb543269c1eca3d841f5e | 0 | −49 | 0 | 0 |
| 0x0a2fbc9aa7f683ac59e15a80367f0893203cb2b4 | 5 | −0.5 | 0 | 0 |
| 0x0a4e1a270c7aaad91afbbf55fdc0e32c4771a98d | 6 | −2 | 0 | 0 |
| 0x0a4ff45da57390fb3a98f06a18dfd767898b9e1f | 7 | −6 | 0 | 0 |
| 0x0ef22f9a5bd6e89554365b337c1c263d5e15920e | 0 | −39 | 0 | 0 |
| 0x0f1366a597fa7ec9b6425fe716a63aaaf3eb60a1 | 2 | −1 | 0 | 0 |
| 0x0f3667a010252ea3ef043e9c63fe23d6d557d4a8 | 3 | −5 | 0 | 0 |
| … | … | … | … | … |
| 0x12aaf9107b4e96f972fc920bf3bd0e651fc8bcd5 | 35 | 33.25 | 0 | 0 |
| 0xdcf37d8aa17142f053aaa7dc56025ab00d897a19 | 438 | −122 | 0 | 1 |

can see that there are 18 pre-set malicious nodes that are detected as malicious nodes through our model. The accuracy of this test was ninety per cent. The results show that our proposed method is effective in malicious node detection.

## CONCLUSIONS

With the development of SIoT systems, the emergence of consortium blockchain provides a new secure exchange platform for SIoT's valuable data. A distributed environment is the common feature of the consortium blockchain and SIoT. There are many trust problems in a distributed environment, and malicious node judgment can improve the credibility of the whole system. This article mainly aims at the users who pass the consortium blockchain node, analyzes the operation times of these users deployed to the consortium blockchain as the primary data, and then analyzes the processing by the specific processing delay algorithm. A machine learning clustering algorithm is used to distinguish between malicious nodes and normal nodes by dividing time zones. The results show that the density-based clustering algorithm DBSCAN can distinguish them well.

In future research, we can detect specific attacks by analysing the specific behavioural patterns of the nodes. At the same time, we can compare and analyse data from multiple time periods to gain more insight into the patterns of malicious nodes. Further processing of malicious node data such as specific access control methods or specific trust algorithms can also be performed. Malicious and normal nodes will be processed through detailed trust algorithms. By introducing trust scoring to score nodes in both directions, the scoring system will make the system safer and more stable to judge malicious nodes and improve the whole system. This also provides a good security foundation for future SIoT so that SIoT can achieve continuous and stable development.

### Funding

This work was supported by the National Natural Science Foundation of China under Grant 61872051. The funders had no role in study design, data collection and analysis, decision to publish, or preparation of the manuscript.

### Grant Disclosures

The following grant information was disclosed by the authors:
National Natural Science Foundation of China: 61872051.

### Competing Interests

The authors declare that they have no competing interests.

### Author Contributions

- Song Luo conceived and designed the experiments, performed the computation work, prepared figures and/or tables, authored or reviewed drafts of the article, and approved the final draft.
- Lianghai Lai conceived and designed the experiments, performed the experiments, performed the computation work, prepared figures and/or tables, authored or reviewed drafts of the article, and approved the final draft.

- Tan Hu performed the experiments, analyzed the data, authored or reviewed drafts of the article, and approved the final draft.
- Xin Hu analyzed the data, authored or reviewed drafts of the article, and approved the final draft.

## Data Availability

The raw data and code are available in the Supplemental Files.

## Supplemental Information

Supplemental information for this article can be found online at http://dx.doi.org/10.7717/peerj-cs.2108#supplemental-information.

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
