# Peer review of "Detection of malicious nodes based on consortium blockchain"

_PeerJ Computer Science, doi:10.7717/peerj-cs.2108_

## Round 0.1 · original submission · Major Revisions

The paper proposes a method for detecting malicious nodes in blockchain-based Social Internet of Things, mainly using inter-block delay features. However, citation formatting issues and concerns regarding clarity, proofreading, and relevance of Figure 2 arise. While the authors simulated transactions and utilized transaction processing time data to train machine learning models, the methodology for simulating attacks for evaluation lacks clarity. Furthermore, the paper lacks clear explanations on how the detection method relates to various types of attacks and justification for its subjectivity and uncertainty. The Related Work section lacks logical structure. Addressing these concerns is crucial for enhancing clarity and validity, thus maximizing the paper's contribution to the field.

**Language Note:** The review process has identified that the English language must be improved. PeerJ can provide language editing services - please contact us at copyediting@peerj.com for pricing (be sure to provide your manuscript number and title). Alternatively, you should make your own arrangements to improve the language quality and provide details in your response letter. – PeerJ Staff

Reviewer 1 ·

Basic reporting

The authors do not format the citations in a proper way, the citations in this paper is not numbered. Please correct the references using the correct format.

Experimental design

The authors simulated some transactions on a consortium chain they built. They then used those transaction processing time as the data and tried out several machine learning models to distinguish whether a node is malicious or not.

Validity of the findings

N/A

Additional comments

Thanks for submitting your paper. This paper tries to solve an interesting and important problem on blockchain -- detecting malicious nodes in a decentralized environment. However, I have several concerns as follows.

[Formatting issues] The authors are not citing papers in a proper way. Please fix it.

[Intuition about definition of malicious nodes] The authors seem to claim in this paper that malicious nodes are going to raise intentional delay to block the entire system. However, in my opinion, this claim is not well justified. For example, if a node is running on a machine with very poor computing resources, so it computes everything slowly. Do such delays really constitute a malicious node? I wish the authors could provide more reasoning on why the definition of malicious node is reasonable.

[Validity of the dataset] As I understand it, the authors simulate transactions of all the nodes and feed all these input data to the machine learning model. However, I don't believe such simulation is representative enough. The dataset will only capture the defined malicious behaviors, but in real world, malicious nodes can do more than just delaying services. I indeed have concerns about the validity of the dataset thus I am not fully convinced that the proposed methods are effective as the authors stated.

Cite this review as

Reviewer 2 ·

Basic reporting

This paper presents a method for detecting malicious nodes in the blockchain-based Social Internet of Things (SIoT). The detection method primarily utilizes the inter-block delay features of nodes in the consortium blockchain. It employs unsupervised clustering algorithms, such as K-means and DBSCAN, to train detection models.
My main concerns are as follows.
(1) I suggest changing "consortium chain" to "consortium blockchain," as it is a commonly used term in the blockchain domain.
(2) A thorough proofreading is necessary to rectify errors regarding the use of commas and periods.
(3) Figure 2 occupies an entire page but contributes very little to the paper.

Experimental design

(4) In the evaluation section, a simulated dataset is generated and used to assess performance. However, it is unclear how the attacks are simulated. Please provide a detailed explanation of the methodology used to simulate the attacks for evaluation purposes.

Validity of the findings

(5) The goals of the detection method are currently unclear and confusing. The statement "the method is subjective, uncertain, lightweight, and dynamically time-sensitive for detecting malicious nodes" lacks clarity. It is not apparent why this detection method should be "subjective and uncertain." Please provide further elaboration to clarify the objectives of the method.

(6) The Related Work section lacks logical structure. It would be better to divide the paragraph into several logical sections. Each section should summarize and analyze a specific type of existing work separately.

(7) The paper primarily utilizes the delay feature to detect malicious nodes. However, it is unclear how this relates to the mentioned attacks, such as data tampering, negative transmission, denial of service, attacks on other nodes, and malicious access to data. Provide a clear explanation of the relationship between delay and these attacks, as well as the observations that led to the proposal of this detection method.

Additional comments

N/A

Cite this review as

---

## Round 0.2 · Major Revisions

The revised draft shows commendable efforts in addressing formatting issues and previous reviewers' concerns. However, the evaluation methodology's effectiveness remains a major concern due to the lack of ground-truth data for identifying malicious nodes, undermining the credibility of the proposed methods. It is crucial for the authors to reference established literature or acknowledged methods to determine truly malicious nodes and compare them with nodes identified by their proposed methods to substantiate their contribution effectively. Additionally, removing Fig. 2, breaking down paragraphs in the Related Work section for enhanced readability, and providing further clarification on the dataset's rationale and observed attacks will strengthen the manuscript's coherence and relevance. Incorporating these improvements is pivotal for the paper's acceptance and impact in the field.

Reviewer 2 has requested that you cite specific references. I do not expect you to include these citations.

Reviewer 1 ·

Basic reporting

The authors fixed citation format issues.

Experimental design

N/A

Validity of the findings

N/A

Additional comments

Thanks for submitting your revised draft. And I appreciate the efforts that the authors have put into addressing the previous reviewers' concerns and fixing the formatting issues.

After reviewing the response and revised draft, I still have several major concern which hinder my assessment of this work. While authors admit that the methods of defining malicious nodes is subjective and is based on the fact the malicious nodes' historical transaction data may vary very much, the major concern is that the evaluation lacks the ground-truth data.

In the evaluation the authors used several clustering methods to classify nodes into malicious or normal. However, as there are no ground-truth malicious nodes data (which means that the nodes are indeed malicious classified through some metrics), just labeling the nodes is not convincing to show that the proposed methods are effective or not.

I suggest that the authors look for some literature or commonly acknowledged methods to determine which nodes are really malicious. Then in the evaluation, the authors should compare the malicious nodes detected via the proposed methods with the ground-truth data. This is essential to really understand the contribution of the work.

Cite this review as

Reviewer 2 ·

Basic reporting

I would like to express my gratitude to the authors for their efforts in revising the manuscript. I am pleased to see that most of my previous concerns have been addressed. However, I still have a few comments that I would like to share:

(1) I recommend removing Fig. 2, as it seems to only present some application scenarios of blockchain. The information conveyed in this figure can be adequately explained using words only.

(2) I suggest breaking down the second paragraph of the Related Work section into shorter, more concise paragraphs. This will greatly enhance the overall readability of the section.

(3) I encourage the authors to include references to additional related works on consortium blockchain, such as:

a) "On Private Data Collection of Hyperledger Fabric" - presented at ICDCS 2021.
b) "On Security of Proof-of-Policy (PoP) in the Execute-Order-Validate Blockchain Paradigm" - presented at CNS 2022.
c) "BBS: A Blockchain Big-Data Sharing System" - presented at ICC 2022.
d) "CrowdBC: A blockchain-based decentralized framework for crowdsourcing" - published in TPDS 2018.

Experimental design

(4) Regarding the dataset, the authors mention that they utilize real-world Ethereum transactions. It would be beneficial to provide further explanation regarding the specific attacks observed in the dataset. Additionally, since this paper focuses on consortium blockchain, it would be helpful to clarify the rationale behind using data from a public blockchain like Ethereum.

Validity of the findings

N/A

Additional comments

N/A

Cite this review as

---

## Round 0.3 · Minor Revisions

The authors' commendable efforts to improve their work by providing explanations for malicious node behaviors in experiments are appreciated; however, concerns arise regarding the lack of early acknowledgment of limitations, the absence of citations for mentioned attacks in the "Comparative Analysis of Experiments" section, and the need for justification or larger-scale experiments given the limited scale of 100 nodes.

Reviewer 1 ·

Basic reporting

The authors fixed the formatting issues and some grammatical issues.

Experimental design

Authors explained how the malicious nodes are defined and how the data is generated.

Validity of the findings

See below.

Additional comments

I wanted to appreciate the authors' efforts into improving their work. The explanations of which malicious nodes' behaviors are included in your experiments helped better assessment to the experiments. However, I still have several concerns and comments for the authors.

1. Please add a paragraph of the limitation of your work early in the introduction to acknowledge that the malicious nodes are only defined by specific kinds of attacks.

2. Please add citations for each kind of malicious node attacks mentioned in "Comparative Analysis of experiments" section.

3. The current experiment is only small-scaled (only simulated behaviors of 100 nodes) which is way much smaller compared to the real-world scenarios. I suggest that the authors conduct larger-scale experiments with more nodes to demonstrate the effectiveness of your methods. Or please explain why 100 nodes are enough and include this in the limitation paragraph.

Cite this review as

Reviewer 2 ·

Basic reporting

The authors have addressed my previous concerns. I do not have further comments.

Experimental design

The authors have addressed my previous concerns. I do not have further comments.

Validity of the findings

The authors have addressed my previous concerns. I do not have further comments.

Additional comments

The authors have addressed my previous concerns. I do not have further comments.

Cite this review as

---

## Round 0.4 · accepted · Accept

Congratulations! Thanks for the revision!

Reviewer 1 ·

Basic reporting

N/A

Experimental design

N/A

Validity of the findings

N/A

Additional comments

I want to thank the authors for submitting their revised draft. They cleared my concern. Just one minor request for the final camera ready version (but I am not expecting the authors to submit another revision), please try to include your comments on why 100 nodes are enough in your paper.

Cite this review as